# Improving classifier decision boundaries and interpretability using nearest neighbors

## Abstract

Neural networks are not learning optimal decision boundaries. We show that decision boundaries are situated in areas of low training data density. They are impacted by few training samples which can easily lead to overfitting. We provide a simple algorithm performing a weighted average of the prediction of a sample and its nearest neighbors' (computed in latent space) leading to minor favorable outcomes for a variety of important measures for neural networks. In our evaluation, we employ various self-trained and (state-of-the-art) pre-trained convolutional neural networks to show that our approach improves (i) resistance to label noise, (ii) robustness against adversarial attacks, (iii) classification accuracy, and yields novel means for (iv) interpretability. Our interpretability analysis is of independent interest to the XAI community, as it is applicable to any network. While improvements are not necessarily large in all four areas, our approach is conceptually simple, i.e., improvements come without any modification to network architecture, training procedure or dataset. Furthermore, our approach is in stark contrast to prior works that often require trade-offs among the four objectives combined with architectural adaptations or provide valuable, but non-actionable insights. Finally, we provide a theoretical analysis.

## 1 Introduction

In the realm of machine learning, the decision boundary plays a crucial role in distinguishing between classes. Classes typically share certain characteristics and tend to form clusters. The decision boundary is a hypersurface that partitions the input space into regions corresponding to different classes. An optimal decision boundary implies an optimal classifier and vice versa. Simple classifiers, such as support vector machines (SVMs), logistic regression, and k-nearest neighbors, often generate simple decision boundaries (Cortes & Vapnik, 1995). On the other hand, deep neural networks (DNNs) have shown remarkable capabilities in learning feature hierarchies and capturing complex, non-linear decision boundaries owing to their depth and non-linear activation functions (LeCun et al., 2015). They can be said to have revolutionized the field of computer vision and others, leading to astonishing improvements in accuracy on multiple benchmarks. Still, these models also suffer from weaknesses such as a lack of interpretability, lack of robustness as witnessed by the effectiveness of adversarial samples (Goodfellow et al., 2014). Many techniques that tackle the problem of adversarial samples, interpretability, as well as improving handling of noisy labels come with tradeoffs. That is, the pursuit of any of these goals often leads to lower accuracy or requires altering training schemes, datasets, and architectures.

In this work, we propose a technique that should tackle all of these issues as illustrated in Table 1. We combine the prediction of a pre-trained neural network of a sample to predict and the prediction of the k-nearest neighbors (kNNs). We compute nearest neighbors (NNs) in latent space using layer activations of a (pre-trained) classifier. We calculate a weighted average of the actual prediction and those of NNs as final prediction (see Figure 1).

This approach improves robustness against adversarial samples, interpretability, and handling noisy samples without compromising performance of the classifier, i.e., we mostly improve it. It also does not require altering a (pre-trained) classifier. However, obtaining kNNs is also computationally expensive and our technique does also not fully address the desiderata (in Table 1), but rather it marks a step forward. This, is still a major achievement given that other techniques require rather

Figure 1: Method: The training data and the sample to infer is run through the classifier up to some layer yielding an embedding to compute the kNNs. A weighted average of the last layer of the sample to infer and the NNs is used for prediction.

unpleasant trade-offs. Our work is also interesting as it sheds new insights on decision boundaries and interpretability based on training data that are applicable even when kNNs are not employed for predictions. We show that classifiers do generally not learn optimal decision boundaries (also) due to the fact that these boundaries lie in areas of few training samples and thus naturally suffer from overfitting. Predictions of samples near the decision boundary, i.e., samples in these sparse areas, can benefit from using NNs. While we expect few (test) data in such sparse areas, differences when altering the boundary using NNs can still be observed. Our work also allows a novel form of simple contrastive analysis by focusing on instances, where adding NNs actually caused a change in prediction. As shown in our evaluation, this allows more easily to hypothesize what characteristics are responsible for a decision and, thus, contributes to the field of explainability(XAI)(Longo et al., 2024).

Table 1: Comparison to prior work

| Method | Improvements in | | | | Can use pretrained networks? | Better understanding of decision boundary? |
| | Accuracy | Interpretability | Advers. Robust. | Label Noise Robustness | | |
| --- | --- | --- | --- | --- | --- | --- |
| Wu et al. (2020) | | | | ✓ | | ✓ |
| Ortiz-Jimenez et al. (2020) | | | | | | ✓ |
| Karimi et al. (2019) | | | | | | ✓ |
| Yang et al. (2020) | | ✓ | | | | ✓ |
| Oyen et al. (2022) | | | | | | ✓ |
| Papernot & McDaniel (2018) | (✓) | ✓ | ✓ | | | |
| **Ours** | ✓ | ✓ | ✓ | ✓ | ✓ | ✓ |

## 2 METHOD

Our method is different from classical kNNs in three ways. First, we compute nearest neighbors based on a latent representation given by layer activations rather than on input samples or final outputs. Second, the prediction is a combination of the network output of the sample to predict and its NNs, while for classical kNNs only the NNs are used to make a prediction. Third, the combination is based on directly aggregating network outputs rather than performing a majority vote of the classes of the kNNs.

---
**Algorithm 1** LAtent-SElf-kNN (LaSeNN)
---
1: **Input:** Classifier $C$, (training) data $\mathcal{D}$, sample to predict $x_q$
2: **Output:** Prediction $y_p$
3: $k := 3$ ▷ number of NNs
4: $w := 0.88$ ▷ weight of sample $x_q$
5: $j := n - 2$ ▷ Layer index to obtain embeddings used for similarity computation for NNs
6: $sim(x, x') := ||x - x'||_2^2$ ▷ similarity metric for NN
7: $L_D := \{(C_{:j}(x_i), y_i)|(x_i, y_i) \in D\}$
8: $NN_k(C_{:j}(x_q)) :=$ $k$-NNs of $C_{:j}(x_q)$ in $L_D$ using metric $sim$
9: $x_w^j = w \cdot C(x_q) + (1 - w) \cdot \frac{\sum_{x \in NN_k} C(x)}{k}$
10: $y_p := \arg\max_o x_{w,o}^j$ ▷ Predicted class is index $o$ of "neuron" with maximal output
---

The method is illustrated in Figure 1. More formal pseudocode is shown in Algorithm 1 called "LAtent-SElf-kNN"(LaSeNN), since it combines the sample to predict and its NNs computing NNs

based on similarity in latent space. We are given a classifier $C = (L_1, ..., L_N)$ consisting of $N$ layers, a training dataset $\mathcal{D} = \{(x_i, y_i)_{i=1}^n\}$ and a query sample $x_q$ used for inference. We compute the activations $C_{:j}(x_i)$ of layer $j$ for all samples in the training dataset and the sample for inference, i.e. $x \in \{\mathcal{D} \cup \{x_q\}\}$. Then we compute the $k$-nearest neighbors $NN_k(C_{:j}(x_q)) \subset \mathcal{D}$ and, finally, a weighted average

$$C(x_w) = w \cdot C(x_q) + (1 - w) \cdot \frac{\sum_{x \in NN_k} C(x)}{k}$$

The underlying motivation is illustrated in Figure 3. Classes form dense clusters that are separated by sparse space. The decision boundary runs through the sparse space. The exact location of the boundary is heavily influenced by the samples in the sparse space and it likely overfits. Generally, using NNs can lead to a smoother boundary that is simpler and less-prone to overfitting (see e.g. textbooks like Hastie et al. (2009)).

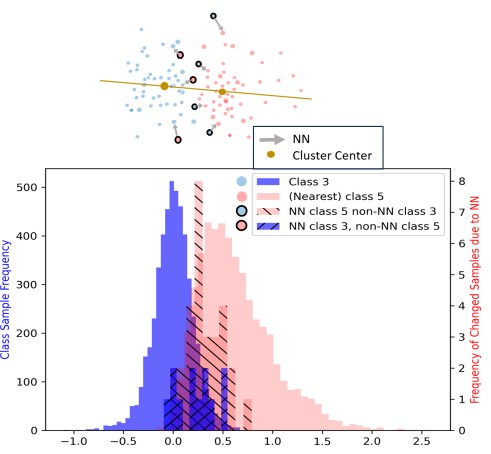

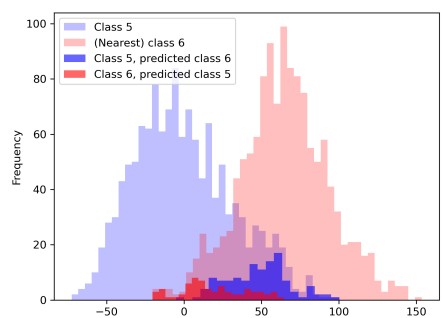

(a) Illustration of points of two classes above. The histogram below shows the distribution of (length of) projections onto the line connecting the cluster centers for a VGG-13 trained on Cifar-10 using the third last convolutional layer. It shows that NNs of wrongly classified points are more often of the correct class than not.

(b) Changes of predictions due to Algorithm LaSeNN illustrated using the distribution of projections (see Figure 2a – top panel) for a ResNet-34 trained on Imagenet using the layer (outputs) prior to the last dense layer. It illustrates that changes occur mostly in lower density areas.

Figure 2: Comparison of VGG-13 on Cifar-10 and ResNet-34 on Imagenet using Nearest Neighbor-based analysis.

Predictions for samples near any of the cluster centers are relatively far from the decision boundary and bear little uncertainty and are likely correct. They are not impacted by our method, i.e., the prediction using Algorithm 1 (LaSeNN) and the prediction of the network without using NNs is identical. However, predictions in the sparse space are potentially close to the decision boundary, which is strongly influenced by a few samples. Using kNNs leads to changes to the boundary in this space, i.e., a 'novel' decision boundary as illustrated conceptually in Figure 3. (Actual data is shown in Figures 2a and 2b.) As shown in Figure 2a, for a class $c_0$ we compute the mean of class samples and find the class $c_1$ with the mean that is at minimum L2-distance. We then compute the projection of points $x$ onto the line connecting the centers, i.e. $(x - c_0) \cdot (c_1 - c_0)$ (based on an ordinary dot product) and create histograms of the length of the projection. Figure 2a shows that errors, i.e.,"confusion" of the two classes occurs primarily in areas of lower density. Figure 2b shows that changes of samples due to Algorithm LaSeNN also occur primarily in low density areas. Furthermore, there are only relatively few changes, e.g., the dense areas containing most samples are not impacted by our method, but only areas of low density. Note that Figure 2b has a twin axis, i.e., the left y-axis is for the distribution of class samples counts and the right one only for those samples which prediction got changed due to the use of NNs, i.e. Algorithm LaSeNN.

The returned NNs help to better understand classifier decisions, i.e., they are well-interpretable. They indicate which samples of the training data contribute at least the fraction $(1 - w)$ to the decision, i.e., each classifier output of a NN has a weight of $\frac{1-w}{k}$. If the layer $i$ is close to the output, the NNs also resemble samples that are considered "very similar" by the classifier and therefore

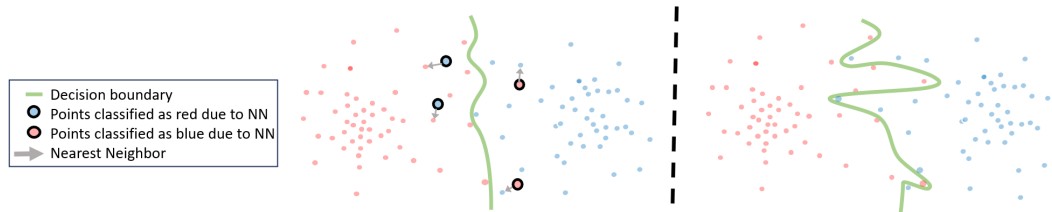

Figure 3: The decision boundary between two classes (solid green line) is influenced by a few points in a sparse area. Using NNs (left panel) might lead to a smoother boundary in the sparse area (compare left and right panel).

can help in understanding, which concepts are relevant (see concept-based XAI techniques such as Schneider & Vlachos (2022)). This can be leveraged in particularly, when class predictions change due to NNs as discussed in our evaluation.

## 3   EVALUATION

Our evaluation focuses on image classification using various classifiers and datasets. We do the following:

- Assessing basic assumptions on distribution of layer activations (Section 3.2)

- Analyzing parameter sensitivity, e.g., impact of number of NNs and their weight on classifier accuracy (Section 3.3)

- Robustness to adversarial samples (Section 3.5) and label noise (Section 3.4)

- Discussing interpretability focusing on changes to predictions due to NNs (Section 3.6)

- Performance of Algorithm LaSeNN for unchanged, pre-trained classifiers on ImageNet (Section 3.7)

### 3.1   DATASETS, NETWORKS AND SETUP

Datasets used are CIFAR-10/100 (Krizhevsky & Hinton, 2009), scaled to 32x32, and ImageNet (Deng et al., 2009). As networks we used VGG (Simonyan & Zisserman, 2014), Resnet (He et al., 2016), MobileNetv3 (Howard et al., 2019) and ConvNext (Liu et al., 2022) networks. We used pre-trained networks from the Pytorch's torchvision library v0.15 based on ImageNet (Deng et al., 2009) 'IMAGENET1KV1'. We trained multiple models on CIFAR-10/100 on our own. Training was standard, i.e., stochastic gradient descent with momentum 0.9, batchsize 128, weight decay of 0.0005, no data augmentation and 80 epochs (starting from learning rate 0.11 and decaying it twice by 0.1). We trained five networks for each configuration, i.e. hyperparameter setting and report the mean and standard deviation of metrics. We employ two common targeted adversarial attacks using the Pytorch Advertorch library with default parameters and targets being set to "(index of ground truth class +1) modulo numberOfClasses". Specifically, we use the PGD attack (Kurakin et al., 2016) and the Basic Iterative Attack(BIA) (Madry et al., 2018) which is an iterative extension of FGSM (Goodfellow et al., 2014). If not stated differently, we use Algorithm 1 LaSeNN with the stated parameters in the algorithm. For VGG-13, the third last convolutional layer as layer $i$ by default, while for ResNet-10 we use the output of the second last 'BasicBlock'.

### 3.2   DISTRIBUTION OF LAYER ACTIVATIONS

We aim to assess our assumption that layer activations of (most) samples of one class are closer to each other than to those of other classes, or, put differently, (activations of) class samples form clusters with dense centers that get increasingly sparse towards their boundary, i.e., Figure 3 shows roughly Gaussian shape for the distribution of projections for each class. To verify this assumption, we compute for each point $x$ of the test set, the nearest neighbors $NN_k(x)$ (for $k = 3$) in the training

Table 2: Results for accuracy and adversarial att. on ImageNet

| Net | $corr(P, avgL2)$ | $samePred$ | $avgL2_{corr}$ | $avgL2_{wrong}$ | $avgL2_{change}$ |
|---|---|---|---|---|---|
| ResNet-34 | -0.35 | 0.989 | 3.985 | 4.099 | 4.152 |
| CoNexT-Tiny | -0.49 | 0.993 | 1.434 | 1.565 | 1.663 |
| MobileNetv3-Large | -0.29 | 0.988 | 3.588 | 3.614 | 3.668 |

Table 3: Results for similarity metrics

| Net | Data | Metric | Acc. LaSeNN | Acc. Original | $\Delta$ Acc |
|---|---|---|---|---|---|
| ResNet-10 | Cifar-10 | L2 | 0.855±0.003 | 0.854±0.002 | 0.001±0.001 |
| | | Cosine | 0.854±0.002 | 0.852±0.002 | 0.002±0.0 |
| VGG13 | Cifar-10 | L2 | 0.819±0.002 | 0.816±0.001 | 0.003±0.001 |
| | | Cosine | 0.825±0.001 | 0.816±0.001 | 0.008±0.001 |
| ResNet-10 | Cifar-100 | L2 | 0.579±0.001 | 0.574±0.002 | 0.004±0.001 |
| | | Cosine | 0.585±0.003 | 0.575±0.002 | 0.01±0.002 |
| VGG13 | Cifar-100 | L2 | 0.511±0.003 | 0.505±0.002 | 0.006±0.0 |
| | | Cosine | 0.522±0.003 | 0.505±0.002 | 0.017±0.001 |

dataset and compute (1) pureness $P$: number of samples within $NN_k(x)$ that are of the same class as $x$ and (2) the average L2-distance $avgL2$ of the NNs to $x$[1].

If our assumption is correct, we expect that density measured by average $L2$ distance ($avgL2$) and pureness $P$ are negatively correlated $corr(P, avgL2) < 0$, i.e., higher density is expected for points of the same class (high pureness) and lower density for points of distinct classes(low pureness). As shown in Table 2 the Pearson correlation yielded values between -0.29 to -0.49 for all pretrained networks with p-values $< 0.001$. We also expect that most predictions remain unaltered due to using Algorithm LaSeNN, which is confirmed in Table 2 showing that more than 98% of all samples yield the same class prediction ($samePred$) if we compare the predictions of LaSeNN and the native classifier. We also expect that the mean distance to neighbors is lower for correctly classified points (since they are in dense areas near a center of a class) than for incorrectly classified samples (since they are in sparser areas with samples of different classes), i.e., $avgL2_{corr} < avgL2_{wrong}$, which is also confirmed (see Table 2). We also expect that changes of class predictions due to LaSeNN occur primarily in low density areas (e.g. for large mean distances), i.e., $avgL2_{change} > avgL2$, which is also confirmed.

### 3.3 PARAMETER SENSITIVITY

First, we assess the sensitivity to similarity metrics, i.e., we evaluate two common similarity metrics for high dimensional vectors: the (negative) $L2$-norm $sim(x, x') = -||x - x'||_2^2$ and cosine similarity $sim(s, s') = \text{cosine}(x, x')$. Our evaluation shows that both lead to gains when using NNs but cosine leads to larger gains. This is expected since the space is relatively sparse (the Cifar 10/100 datasets are small with just 50k samples and the number of dimensions is large (at least 512 dimensions). In sparse spaces measures like cosine that neglect magnitude and are only concerned with direction are more favorable. In turn, L2 is more adequate for dense spaces, i.e., large training datasets – We used L2 for our benchmarks with Imagenet.

The outcomes for different layers $i$ are shown in Figure 4. Using layers further from the output though not too close to the input tends to lead to best results. These layer outputs still maintain a significant amount of information about the input, i.e., are not too much tuned to be discriminative, while still capturing task specific aspects – as can be seen when comparing reconstruction and classification losses (Schneider & Prabhushankar, 2024). Given a network of sufficient capacity all training samples will be perfectly classified after training, meaning they will have close to zero loss. In turn, the output of the very last layer is similar for all samples of a class, e.g., samples cannot be well discriminated using the final layer output.

In Table 5 we see that using NNs in addition to the sample to classify leads to gains from 0.1% up to about 3%. Gains are largest if the weight $w$ of the sample to predict is about 75% and that of the neighbors jointly only 25% though there is no strong sensitivity of the weight $w$. Using only NNs

---

[1]Distance to the kNN has been employed for density-based clustering, e.g., Schneider & Vlachos (2017)

Table 4: Results for layer $i$ used for similarity computation

| Net | Data | Layer $i$ | Acc. LaSeNN | Acc. Original | $\triangle$ Acc |
|---|---|---|---|---|---|
| ResNet-10 | Cifar-10 | prior to dense | 0.853±0.002 | 0.852±0.002 | 0.0±0.0 |
| | | prior to 4x4 pool | 0.854±0.002 | 0.852±0.002 | 0.002±0.0 |
| | | prior to last block | 0.864±0.004 | 0.854±0.002 | 0.01±0.002 |
| VGG13 | Cifar-10 | prior to dense | 0.817±0.001 | 0.816±0.001 | 0.001±0.0 |
| | | 2nd last conv | 0.816±0.001 | 0.816±0.001 | -0.001±0.001 |
| | | 4th last conv | 0.825±0.001 | 0.816±0.001 | 0.008±0.001 |
| | | 6th last conv | 0.824±0.001 | 0.816±0.001 | 0.008±0.002 |
| ResNet-10 | Cifar-100 | prior to dense | 0.581±0.001 | 0.574±0.002 | 0.007±0.002 |
| | | prior to 4x4 pool | 0.585±0.003 | 0.575±0.002 | 0.01±0.002 |
| | | prior to last block | 0.593±0.001 | 0.574±0.002 | 0.019±0.001 |
| VGG13 | Cifar-100 | prior to dense | 0.508±0.002 | 0.505±0.002 | 0.003±0.001 |
| | | 2nd last conv | 0.518±0.002 | 0.505±0.002 | 0.012±0.0 |
| | | 4th last conv | 0.522±0.003 | 0.505±0.002 | 0.017±0.001 |
| | | 6th last conv | 0.518±0.003 | 0.505±0.002 | 0.013±0.001 |

Table 5: Results for weight $w$

| Net | Data | $w$ | Acc. LaSeNN | Acc. Original | $\triangle$ Acc |
|---|---|---|---|---|---|
| ResNet-10 | Cifar-10 | 0 | 0.853±0.0 | 0.852±0.0 | 0.001±0.0 |
| | | 0.52 | 0.854±0.0 | 0.852±0.0 | 0.001±0.0 |
| | | 0.76 | 0.854±0.0 | 0.852±0.0 | 0.002±0.0 |
| | | 0.88 | 0.854±0.002 | 0.852±0.002 | 0.002±0.0 |
| | | 0.94 | 0.854±0.0 | 0.852±0.0 | 0.002±0.0 |
| | | 0.97 | 0.853±0.0 | 0.852±0.0 | 0.001±0.0 |
| VGG13 | Cifar-10 | 0 | 0.799±0.002 | 0.816±0.001 | -0.017±0.004 |
| | | 0.52 | 0.825±0.001 | 0.816±0.001 | 0.009±0.002 |
| | | 0.76 | 0.831±0.0 | 0.816±0.001 | 0.015±0.001 |
| | | 0.88 | 0.825±0.001 | 0.816±0.001 | 0.008±0.001 |
| | | 0.94 | 0.821±0.001 | 0.816±0.001 | 0.004±0.001 |
| | | 0.97 | 0.819±0.001 | 0.816±0.001 | 0.002±0.001 |
| ResNet-10 | Cifar-100 | 0 | 0.586±0.0 | 0.577±0.0 | 0.01±0.0 |
| | | 0.52 | 0.586±0.0 | 0.577±0.0 | 0.009±0.0 |
| | | 0.76 | 0.587±0.0 | 0.577±0.0 | 0.01±0.0 |
| | | 0.88 | 0.585±0.003 | 0.575±0.002 | 0.01±0.002 |
| | | 0.94 | 0.585±0.0 | 0.577±0.0 | 0.008±0.0 |
| | | 0.97 | 0.58±0.0 | 0.577±0.0 | 0.004±0.0 |
| VGG13 | Cifar-100 | 0 | 0.448±0.003 | 0.505±0.002 | -0.058±0.004 |
| | | 0.52 | 0.525±0.002 | 0.505±0.002 | 0.02±0.002 |
| | | 0.76 | 0.536±0.002 | 0.505±0.002 | 0.031±0.002 |
| | | 0.88 | 0.522±0.003 | 0.505±0.002 | 0.017±0.001 |
| | | 0.94 | 0.515±0.004 | 0.505±0.002 | 0.01±0.002 |
| | | 0.97 | 0.511±0.002 | 0.505±0.002 | 0.005±0.0 |

(instead of the sample to classify) can be much worse (i.e. for VGG13), but it can also be slightly beneficial (e.g. for ResNet-10).

Considering the number of neighbors $k$ (Table 7), we see that overall improvements are largest, if just a single nearest neighbor is used. This is not surprising, since the space is sparse and, thus, the larger $k$ the more distant and dissimilar the neighbors become and the less valuable they are for prediction, i.e., they are more likely of another class than the ground truth class.

### 3.4 NOISY LABELS

Table 6 shows that using nearest neighbors leads to larger gains with growing noise, i.e., if we permute an increasing fraction of labels in the training data and the classifier is trained on this noisy data. This suggests that in latent space (induced by a classifier layer) training samples with permuted (incorrect) label are still placed near samples of the correct label since they share similarities (beyond the class label).

### 3.5 ROBUSTNESS TO ADVERSARIAL ATTACKS

In Table 8 we see that the difference between LaSeNN and the unmodified classifier is larger for both of the targeted adversarial attacks indicating that our approach increases robustness to adversarial

Table 6: Results for noisy labels

| Net, Data | Perm. labels | Acc. LaSeNN | Acc. Original | Δ Acc |
|---|---|---|---|---|
| ResNet-10 Cifar-10 | 0.0 | 0.854±0.002 | 0.852±0.002 | 0.002±0.0 |
| | 0.01 | 0.841±0.0 | 0.839±0.001 | 0.002±0.0 |
| | 0.04 | 0.807±0.0 | 0.807±0.001 | 0.0±0.0 |
| | 0.08 | 0.77±0.0 | 0.766±0.0 | 0.003±0.001 |
| | 0.16 | 0.708±0.002 | 0.703±0.001 | 0.005±0.001 |
| | 0.32 | 0.59±0.003 | 0.581±0.003 | 0.01±0.0 |
| VGG-13 Cifar-10 | 0.0 | 0.825±0.001 | 0.816±0.001 | 0.008±0.001 |
| | 0.01 | 0.814±0.003 | 0.806±0.004 | 0.008±0.001 |
| | 0.04 | 0.796±0.001 | 0.783±0.002 | 0.013±0.002 |
| | 0.08 | 0.767±0.004 | 0.753±0.004 | 0.014±0.0 |
| | 0.16 | 0.716±0.001 | 0.696±0.001 | 0.02±0.0 |
| | 0.32 | 0.604±0.0 | 0.577±0.003 | 0.026±0.003 |
| ResNet-10 Cifar-100 | 0.0 | 0.585±0.003 | 0.575±0.002 | 0.01±0.002 |
| | 0.01 | 0.576±0.0 | 0.566±0.0 | 0.01±0.0 |
| | 0.04 | 0.542±0.001 | 0.529±0.001 | 0.013±0.001 |
| | 0.08 | 0.502±0.002 | 0.49±0.002 | 0.012±0.0 |
| | 0.16 | 0.437±0.0 | 0.424±0.002 | 0.013±0.002 |
| | 0.32 | 0.332±0.002 | 0.315±0.002 | 0.017±0.0 |
| VGG-13 Cifar-100 | 0.0 | 0.522±0.003 | 0.505±0.002 | 0.017±0.001 |
| | 0.01 | 0.52±0.001 | 0.501±0.001 | 0.019±0.002 |
| | 0.04 | 0.493±0.002 | 0.473±0.003 | 0.02±0.001 |
| | 0.08 | 0.47±0.002 | 0.45±0.002 | 0.02±0.001 |
| | 0.16 | 0.422±0.0 | 0.399±0.001 | 0.022±0.001 |
| | 0.32 | 0.33±0.01 | 0.307±0.007 | 0.024±0.003 |

Table 7: Results for nearest neighbors $k$

| Net, Data | k | Acc. LaSeNN | Acc. Original | Δ Acc |
|---|---|---|---|---|
| ResNet-10 Cifar-10 | 8 | 0.856±0.003 | 0.854±0.002 | 0.002±0.001 |
| | 4 | 0.855±0.003 | 0.854±0.002 | 0.001±0.001 |
| | 3 | 0.854±0.002 | 0.852±0.002 | 0.002±0.0 |
| | 2 | 0.856±0.002 | 0.854±0.002 | 0.002±0.0 |
| | 1 | 0.856±0.002 | 0.854±0.002 | 0.001±0.0 |
| VGG-13 Cifar-10 | 8 | 0.824±0.001 | 0.816±0.001 | 0.007±0.001 |
| | 4 | 0.824±0.001 | 0.816±0.001 | 0.008±0.001 |
| | 3 | 0.825±0.001 | 0.816±0.001 | 0.008±0.001 |
| | 2 | 0.825±0.001 | 0.816±0.001 | 0.009±0.0 |
| | 1 | 0.826±0.001 | 0.816±0.001 | 0.01±0.001 |
| ResNet-10 Cifar-100 | 8 | 0.582±0.002 | 0.574±0.002 | 0.008±0.0 |
| | 4 | 0.584±0.002 | 0.574±0.002 | 0.01±0.0 |
| | 3 | 0.585±0.003 | 0.575±0.002 | 0.01±0.002 |
| | 2 | 0.587±0.002 | 0.574±0.002 | 0.012±0.0 |
| | 1 | 0.587±0.003 | 0.574±0.002 | 0.012±0.001 |
| VGG-13 Cifar-100 | 8 | 0.519±0.003 | 0.505±0.002 | 0.014±0.001 |
| | 4 | 0.522±0.002 | 0.505±0.002 | 0.017±0.001 |
| | 3 | 0.522±0.003 | 0.505±0.002 | 0.017±0.001 |
| | 2 | 0.524±0.002 | 0.505±0.002 | 0.018±0.0 |
| | 1 | 0.527±0.001 | 0.505±0.002 | 0.022±0.001 |

Table 8: Results for adversarial attacks

| Net | Data | Attack | Acc. LaSeNN | Acc. Original | Δ Acc |
|---|---|---|---|---|---|
| ResNet-10 | Cifar-10 | None | 0.854±0.002 | 0.852±0.002 | 0.002±0.0 |
| | | BIA | 0.093±0.01 | 0.09±0.009 | 0.003±0.001 |
| | | PGD | 0.118±0.012 | 0.116±0.012 | 0.003±0.001 |
| VGG13 | Cifar-10 | None | 0.825±0.001 | 0.816±0.001 | 0.008±0.001 |
| | | BIA | 0.235±0.01 | 0.202±0.006 | 0.033±0.006 |
| | | PGD | 0.261±0.011 | 0.237±0.004 | 0.024±0.008 |
| ResNet-10 | Cifar-100 | None | 0.585±0.003 | 0.575±0.002 | 0.01±0.002 |
| | | BIA | 0.045±0.003 | 0.038±0.003 | 0.007±0.001 |
| | | PGD | 0.056±0.003 | 0.048±0.003 | 0.008±0.0 |
| VGG13 | Cifar-100 | None | 0.522±0.003 | 0.505±0.002 | 0.017±0.001 |
| | | BIA | 0.132±0.001 | 0.091±0.002 | 0.04±0.002 |
| | | PGD | 0.15±0.003 | 0.113±0.004 | 0.037±0.005 |

attacks. We believe that this is due to the fact that the adversarial samples are closer to the decision boundary and, thus, are more likely changed, when combined with NNs.

## 3.6 INTERPRETABILITY: HOW DO NNs ALTER PREDICTIONS?

Using NNs for interpretation is not novel, however understanding how they impact predictions to understand models is. That is, we particularly focus on cases, where NNs changed predictions. By investigating what characteristics an input sample and the NNs share, one might gain a better understanding about what aspects a model seems sensitive to, what relevant features for prediction a model seems to lack, and what features it might focus on that might be irrelevant. That is, it can lead to hypothesis that can be investigated using other techniques such as TCAV. For illustration, we used $w = 0.5$ (and $k = 3$), i.e., all three NNs together contribute as much to the prediction as the input sample. Figure 4 shows samples to predict and their NNs, where NNs yielded a change in prediction (more samples are in the Appendix). It is interesting to notice that sometimes NNs can be from different classes although appearing similar, hinting that background has a strong influence. For example, in the first row all NNs differ in class. In the examples, the image is classified as airplane without NNs but correctly as dog using NNs. The first and last NN share partially the typical blue background of airplanes but especially for the first one the object (dog) looks quite different from an airplane. The second image is very different from the predicted and the correct class. Overall the NNs caused a correct prediction suggesting that this could be due to different backgrounds of the NNs (in particular the one of the dog) as well as the second image showing a dog with typical fur textures. To further investigate, one might simply remove any of the NN, adjust

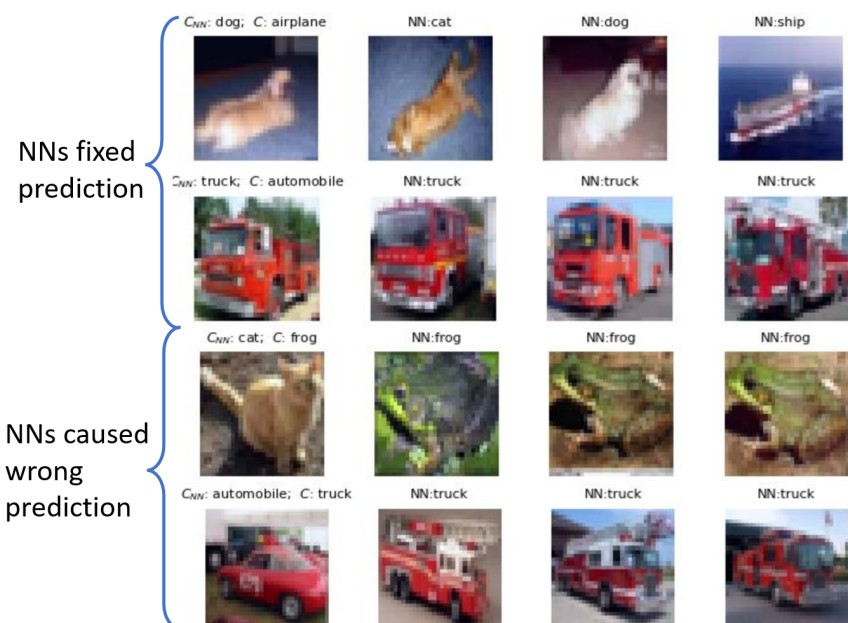

Figure 4: Samples where NN changed the prediction. The first column shows the input to classify, columns 2 to 4 are NNs.

Table 9: Results for accuracy and adversarial att. on pretrained networks on ImageNet

| Net | Attack | Acc. LaSeNN | Acc. Original | $\Delta$ Acc |
|---|---|---|---|---|
| ResNet34 | None | 0.7337 | 0.73316 | 0.00053 |
| | PGD | 0.01189 | 0.00991 | 0.00198 |
| | BIA | 0.01414 | 0.01191 | 0.00222 |
| ConvNext-Tiny | None | 0.82218 | 0.82128 | 0.00090 |
| | PGD | 0.01175 | 0.01035 | 0.00140 |
| | BIA | 0.01388 | 0.01252 | 0.00136 |
| MobileNetV3-Large | None | 0.74154 | 0.74056 | 0.00097 |
| | PGD | 0.00614 | 0.00506 | 0.00108 |
| | BIA | 0.00847 | 0.00707 | 0.00140 |

weights and see if the NN still fixes the prediction. In contrast, in the second row, the trucks appear very similar. However, it can be noted that the misclassified truck as automobile without NNs is somewhat smaller making it more similar to a car. The hypothesis that scale is highly relevant could be further tested by shrinking other samples. For the misclassified samples due to NNs, we see in the last row that a car got misclassified as a truck. The trucks look quite similar, in particular, in the right upper part of the car is a white area (i.e., the top looks like a ladder shown on the trucks) that could contribute to it being classified as truck as the two most similar trucks also have white parts (i.e. a ladder or white colored cabins). Also the perspective of the car is somewhat unusual making the back appear very large. For the second last row, it becomes apparent that quite likely the background (dirt/earth) played a role in obtaining the NN (frogs), leading to a misclassification, which could also be assessed by simple editing the sample to classify.

## 3.7 PRETRAINED NETWORKS

While we have shown accuracy gains and robustness to adversarial samples and label noise on our small self-trained networks, it is unclear to what extent they also exist on large scale networks trained using heavy data augmentation. To this end, we evaluate our technique on multiple pre-trained networks available through Pytorch's Torchvision library using the layer $j$ prior to the last dense layer, $w = 0.94$, and cosine similarity $sim(x, x') = \cosine(x, x')$. We also employ augmentation

for our nearest neighbor query, i.e., we compute the NNs for sample $x_q$ and for the horizontally flipped version $x'_q$ of sample $x_q$. We take the union of the NNs (i.e. the original one and the flipped ones) and take those that are closest. In Table 9 we observe minor gains for all networks. This is somewhat surprising given that all these models are trained based on extensive data augmentation (e.g., random rotation, color jittering, random cropping and resizing, horizontal flipping), while our approach only uses horizontal flipping. Aligned with our self-trained smaller networks we also find that there is an increased robustness to adversarial attacks.

## 4 THEORETICAL ANALYSIS

We analyze a simple scenario to illustrate the potential advantages of integrating neural network predictions with a nearest neighbor approach. While both methods have been in use for decades, theoretical analyses of their hybridization are notably scarce. We only outline key points here and refer to the Appendix for details.

In addition to assuming a one-dimensional data structure, we impose mild assumptions on the data, such as a gradual decrease in density as one moves away from a central region. Establishing results for the general case poses significant challenges due to various interacting factors. These include the non-linearity and high-dimensional parameter spaces inherent in neural networks, the challenge of characterizing the geometry of their decision boundaries (Karimi et al., 2019), and the sensitivity to the underlying data distribution (Section 5). Due to these complexities, we focus our analysis on a critical region $\mathcal{R}$ situated near the decision boundary of the neural network, where classification errors are most likely to occur. The following theorem formalizes our main result.

**Theorem 1** *Under a set of assumptions $S$ specified in Appendix 7, there exists a region $\mathcal{R}$ within the embedding space where the proposed method is expected not to diminish classification performance.*

## 5 RELATED WORK

**Decision boundary**: Studying the decision boundary of neural networks dates back multiple decades (Lee & Landgrebe, 1997; Bishop, 2006). Nowadays, studying the decision boundary is often motivated due to adversarial samples, which show that minor changes to a sample can result in crossing the decision boundary, e.g., deep learning networks are non-robust. Commonly, decision boundaries are also examined using measures and tools found in the context of adversarial examples, e.g., Ortiz-Jimenez et al. (2020); Karimi et al. (2019); Szegedy et al. (2014). Szegedy et al. (2014) discusses adversarial examples in deep learning, illustrating the sensitivity of decision boundaries in neural networks to slight input perturbations. Karimi et al. (2019) generates samples near the decision boundary based on techniques from adversarial samples and in a subsequent step they analyze the generated instances. Nguyen et al. (2015) presents the existence of "fooling" images—unrecognizable inputs that deep neural networks classify with high confidence, highlighting peculiarities in deep learning decision boundaries. Nguyen et al.'s findings shed light on the unusual and unexpected shapes that decision boundaries in deep networks can take. We approach decision boundaries more from the perspective that learnt representations are fixed and the task is to identify an optimal boundary separating samples. Ortiz-Jimenez et al. (2020) leverages tools from adversarial robustness to associate dataset features to the distance of samples to the decision boundary. In turn, they tweak the position of the training samples and measure the resulting changes on the boundaries. They show that deep learning networks exhibit a high invariance to non-discriminative features, and that the decision boundary of a neural network only exist "as long as the classifier is trained with some features that hold them together"(Ortiz-Jimenez et al., 2020).

In addition, there are also a number of theoretical and empirical findings on decision boundaries for neural networks not relying on ideas from adversarial samples. Fawzi et al. (2017) investigates topology of classification regions created by deep networks, as well as their associated decision boundary. The paper claims based on empirical evidence that regions (containing samples of a class) are connected and flat. Li et al. (2018) claims that the decision boundary of the last layer equals that of a hard SVM. Lei et al. (2022) measures the variability of the decision boundary. They show that the more variable the boundary, the less the network generalizes. Recently, Mouton et al. (2023) has predicted generalization performance based on input margins. That is, they use the variability computed based on PCA to assess generalization performance. Nar et al. (2019) argues that

cross-entropy loss leads to poor margins, since samples can be very close to the decision boundary. Support vector machines lead to better margins. In fact, years earlier this has been claimed empirically, i.e., Tang (2013) showed that using a margin-based loss instead of a cross-entropy loss can lead to improvements. Yang et al. (2020) states that thick decision boundaries lead to increased robustness. In the paper they propose training techniques to achieve this, but these techniques lead to significantly worse performance on the clean test sets and only improve on adversarial and out-of-distribution samples.

**Noisy Labels**: The impact of noise on decision boundaries cannot be understated. Noise in the training data can potentially lead to overfitting, manifesting as erratic decision boundaries (Zhang et al., 2021). Large neural networks can "memorize" arbitrary noisy training data (Zhang et al., 2021). However, noisy labels degenerate performance and research has investigated special techniques to deal with label noise. For example, Wu et al. (2020) constructs a topological filter to remove noisy samples. Their approach falls short, when data is non-noisy and it is only shown to yield benefits if a large fraction of labels is noisy. Oyen et al. (2022) showed that label noise depends directly on feature space, i.e.,"when the noise distribution targets decision boundaries, classification robustness can drop off even at a small scale of noise."

**kNN**: Early works (prior to deep learning) (Zhang et al., 2006) trained a SVM on NNs of a query sample. Theoretical works, e.g., Cover (1968), studied also properties of neural networks. However, few theoretical and practical results are known relating deep learning and kNNs. Zhuang et al. (2020) designed a network for training a neural enforcing that a sample and its kNNs all belong to the same class based on a triplet loss. In contrast, we do not constrain training in any way, but rather compute NNs as they emerge by computing them based on the similarity of some layer activation of a trained classifier. Furthermore, our objective is to improve classifiers rather than primarily enforcing that decisions are based (solely) on NNs. Khandelwal et al. (2019) used kNNs for next word prediction. They obtained the kNNs training contexts and computed a distribution of their labels (i.e., words) based on the distance to the test context. Finally, they interpolate the obtained distribution with that of the input sample. We differ in multiple ways: i) we mostly do not use outputs (e.g., outcomes of the softmax layer), but rely on deeper layers; ii) we do not use labels of the training data. Borgeaud et al. (2022); Xu et al. (2023) enhanced LLMs by retrieving contexts based on BERT embeddingsBorgeaud et al. (2022) and RAG Xu et al. (2023). Their work differ as they require architectural changes and a separate embedding model.

**Memory and attention**: Our work also relates to works on including external memory in deep learning (Graves et al., 2016) and to a lesser extent also attention, allowing to focus on specific (input) samples (Vaswani et al., 2017; Bahdanau et al., 2014). Our approach can be said to use training data as a read-only external memory in a static manner, in contrast to differentiable neural computers (Graves et al., 2016) that allow read and write to memory and learn access. Attention allows to attend to inputs within a given sequence. Our approach attends to specific training data used already for network training.

**Explainability**: Explaining using training data is common, e.g., influence functions (Koh & Liang, 2017) allow to explain the impact of training data on decisions. Naively, the influence of a training sample is computed by removing the training sample from the training data retraining the classifier on the reduced data, before assessing how the prediction for a specific sample changes. Our approach does not yield the influence but rather states that the output of a sample is determined by the NNs (at least with fraction $w$). But it is then up to a human to compare the NNs and assess concepts shared among them, e.g., using additional concept-based explainability methods such as TCAV and Schneider & Vlachos (2022).

# 6 CONCLUSIONS

While many approaches exist that target isolated problems such as interpretability, or robustness against label noise, or adversarial robustness, or better performance in general, we achieve with some extra computation"a little bit of most things" using a conceptual simple approach even on pre-trained networks that highlights that "overfitting" is a concern for deep learning on large datasets in areas of low data density. Thus, aside from empirical improvements our work also contributes to a deeper understanding of neural networks.

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

# 7 APPENDIX: THEORETICAL MODEL

We introduce the following set of assumptions, denoted as $S$. First, we utilize a simplified variant of the proposed method that considers only a single nearest neighbor. Second, we assume class balance and no distributional shift between the training and unseen test data, which are standard assumptions in classification tasks. Third, we focus on two classes and assume that the density of points decreases monotonously from a maximum when moving towards the other class, i.e., we use a triangular distribution. Fourth, we assume that the neural network perfectly classifies the training data, a condition that can be attained for sufficiently large networks (Goodfellow et al., 2016). Finally, we assume that within the region $\mathcal{R}$, the classification of a data point by the model is entirely determined by the neural network's classification of its nearest neighbor in the embedding space. It can be realized or approximated in practice by selecting an appropriate weight $w$ and considering that $\mathcal{R}$ is generally small and situated close to the neural network's decision boundary. In such regions, predictions are typically similar and within a narrow range, which makes it easier to find a more stable and less extreme $w$ that prioritizes the nearest neighbor's prediction.

## 7.1 NOTATION

In the following, we assume a suitable underlying probability space with probability measure $\mathbb{P}$ for all probabilistic statements.

For a random variable $X$ taking values in $\mathcal{X}$, we denote its expectation by $\mathbb{E}[X]$ and, when applicable, its probability density function as $f_X$. When referring to a specific realization of a random variable $X$, if possible, we use the corresponding lowercase letter $x \in X$. We also use $\mathbb{1}_A$ to denote the indicator function of the set $A$, i.e., $\mathbb{1}_A(x) = 1$ if $x \in A$ and $\mathbb{1}_A(x) = 0$ otherwise.

## 7.2 PROOF

Consider a training sample $D = \{(x_i, y_i), ..., (x_n, y_n)\}$ where $x_i \in \mathbb{R}$, $n$ is the dimension of input data, and corresponding labels $y_i \in \{blue, red\}$.

Additionally, consider a pre-trained neural network $f$ for binary classification that takes an input $x \in \mathbb{R}$ and outputs the probability that it is classified as 'red'.[2] The network first learns a meaningful data representation, or embedding, and then assigns a label based on this learned representation. For the purpose of our analysis, we simplify this by defining the embedding learned by the network

---

[2]We omit explicit consideration of the neural network weights to keep the notation as simple as possible, as they are not relevant to the subsequent analysis.

for any input $x$ as $z = h(x)$, with the assumption that $z \in [0,1]$. The final layers of the network, which generate the classification output, are represented simply by $\sigma(z)$, thereby abstracting the unnecessary computational details.[3] The binary classification probabilities $\sigma(z)$ are then converted into a proper class variable $\hat{C}_1(z)$ using the following rule:

$$\hat{C}_1(z) = \begin{cases} \text{red} & \text{if } f(x) = \sigma(h(x)) = \sigma(z) \geq 0.5, \\ \text{blue} & \text{otherwise,} \end{cases} \tag{1}$$

where $\sigma$ denotes the sigmoid function.

Next, we investigate the hybrid model introduced in this paper, limited to one nearest neighbor. Specifically, let $z^{(1)}$ denote the nearest neighbor of $z$ in $D$ with respect to the similarity metric chosen.

$$z^{(1)} = argmin_{z' \in \{h(x_i): \, x_i \in D\}} sim(z, z').$$

The label corresponding to $z^{(1)}$ is denoted by $y^{(1)}$.

The hybrid model, predicts the class label $\hat{C}_2(z)$ as follows:

$$\hat{C}_2(z) = \begin{cases} \text{red} & \text{if } w\sigma(z) + (1-w)\sigma(z^{(1)}) \geq 0.5, \\ \text{blue} & \text{otherwise,} \end{cases} \tag{2}$$

where $w$ is a suitable weight within the interval $[0,1]$.

Additionally, as previously stated, we assume that both training and unseen inputs are realizations of the random variables $(X, Y)$. Let $Z = h(X)$ denote the embedding of $X$ generated by the neural network.

After the embedding, data points typically form clusters in the latent space. As specified in the earlier assumptions, to replicate this clustering behavior and simplify the analysis, we model $Z$ given $Y = blue$ as a triangular distribution with parameters $a = 0, b = 1, c = 0$. Similarly, we model $Z$ given $Y = red$ as a triangular distribution with parameters $a = 0, b = 1, c = 1$. Moreover, we assume $\mathbb{P}(Y = blue) = \mathbb{P}(Y = red)$.

Let us now return to the original neural network. We denote its decision boundary in the embedding space as $z^* \in [0,1]$, where $\sigma(z^*) = 0.5$, and we suppose that $z^* < 0.5$.[4] This condition defines a misclassification region for blue points relative to the ideal classifier, represented by:

$$\mathcal{R} := \{z \in [0,1] : \, z \in [z^*, 0.5]\}.$$

As previously specified, however, we assumed that the training data are perfectly classified, though this is not necessarily true for new, unseen data.

In what follows, we show that the proposed hybrid model is expected to maintain or potentially enhance the classification of new inputs $x$ when their embedding $z$ lies within $\mathcal{R}$. This region is particularly important because, as noted earlier, it represents the area where the neural network deviates from an ideal classifier, allowing the hybrid model to provide improvements.

Note that, intuitively, the hybrid algorithm is expected to have a non-negative impact, potentially improving the accuracy in this region. This is because it's more likely to find blue points that were misclassified by the neural network but have a blue neighbor, rather than finding correctly classified red points with a blue neighbor, where the correct classification might be disturbed. To verify this intuition, we will now calculate the difference in expected classification accuracy within the region $\mathcal{R}$ between the hybrid model and the neural network:

$$\Delta Acc_{\mathcal{R}} = \mathbb{E}[\mathbb{1}_{\{\hat{C}_2(Z)=Y\}}(Z,Y) - \mathbb{1}_{\{\hat{C}_1(Z)=Y\}}(Z,Y)|Z \in \mathcal{R}]$$
$$= \mathbb{P}(\hat{C}_2(Z) = Y, \hat{C}_1(Z) \neq Y|Z \in \mathcal{R}) - \mathbb{P}(\hat{C}_2(Z) \neq Y, \hat{C}_1(Z) = Y|Z \in \mathcal{R}).$$

---

[3]It is important to note that the specific functional form assumed here, whether involving a single or multiple final layers, does not impact the subsequent calculations; thus, it is not included among the assumptions stated above for our analysis. The simplification introduced in the text aims to avoid unnecessary details and complications in the subsequent analysis.

[4]This hypothesis is not restrictive; in fact, due to inherent statistical variability in the data, we can reasonably assume that the value will not be exactly 0.5. If instead $z^* > 0.5$, we can simply reverse the roles of the blue and red points in the subsequent analysis.

Observe that,

$$\mathbb{P}(\hat{C}_2(Z) = Y, \hat{C}_1(Z) \neq Y | Z \in \mathcal{R}) = \mathbb{P}(\hat{C}_2(Z) = blue, \hat{C}_1(Z) = red, Y = blue | Z \in \mathcal{R})$$
$$+ \mathbb{P}(\hat{C}_2(Z) = red, \hat{C}_1(Z) = blue, Y = red | Z \in \mathcal{R})$$
$$= \mathbb{P}(\hat{C}_2(Z) = blue, \hat{C}_1(Z) = red, Y = blue | Z \in \mathcal{R})$$
$$= \mathbb{P}(\hat{C}_2(Z) = blue, Y = blue | Z \in \mathcal{R}),$$

where the penultimate and ultimate equality hold because points in $\mathcal{R}$ cannot be classified as blue by the neural network. Analogously, we have:

$$\mathbb{P}(\hat{C}_2(Z) \neq Y, \hat{C}_1(Z) = Y | Z \in \mathcal{R}) = \mathbb{P}(\hat{C}_2(Z) = red, \hat{C}_1(Z) = blue, Y = blue | Z \in \mathcal{R})$$
$$+ \mathbb{P}(\hat{C}_2(Z) = blue, \hat{C}_1(Z) = red, Y = red | Z \in \mathcal{R})$$
$$= \mathbb{P}(\hat{C}_2(Z) = blue, \hat{C}_1(Z) = red, Y = red | Z \in \mathcal{R})$$
$$= \mathbb{P}(\hat{C}_2(Z) = blue, Y = red | Z \in \mathcal{R}).$$

Therefore, $\Delta Acc_{\mathcal{R}}$ becomes:

$$\Delta Acc_{\mathcal{R}} = \mathbb{P}(\hat{C}_2(Z) = blue, Y = blue | Z \in \mathcal{R}) - \mathbb{P}(\hat{C}_2(Z) = blue, Y = red | Z \in \mathcal{R}).$$

Given that $\hat{C}_2(z)$ depends solely on $\hat{C}_1(z^{(1)})$ for $z \in \mathcal{R}$, we can rewrite it as:

$$\Delta Acc_{\mathcal{R}} = \mathbb{P}(\hat{C}_1(Z^{(1)}) = blue, Y = blue | Z \in \mathcal{R}) - \mathbb{P}(\hat{C}_1(Z^{(1)}) = blue, Y = red | Z \in \mathcal{R}).$$

Moreover, since training points are perfectly classified, this difference simplifies to:

$$\Delta Acc_{\mathcal{R}} = \mathbb{P}(Y^{(1)} = blue, Y = blue | Z \in \mathcal{R}) - \mathbb{P}(Y^{(1)} = blue, Y = red | Z \in \mathcal{R}) =$$
$$\mathbb{P}(Y^{(1)} = blue | Y = blue, Z \in \mathcal{R}) \mathbb{P}(Y = blue | Z \in \mathcal{R}) -$$
$$\mathbb{P}(Y^{(1)} = blue | Y = red, Z \in \mathcal{R}) \mathbb{P}(Y = red | Z \in \mathcal{R}).$$

Now, using Byes' theorem we have:

$$\Delta Acc_{\mathcal{R}} = \frac{\mathbb{P}(Y^{(1)} = blue | Y = blue, Z \in \mathcal{R}) \mathbb{P}(Z \in \mathcal{R} | Y = blue) \mathbb{P}(Y = blue)}{\mathbb{P}(Z \in \mathcal{R})} -$$
$$\frac{\mathbb{P}(Y^{(1)} = blue | Y = red, Z \in \mathcal{R}) \mathbb{P}(Z \in \mathcal{R} | Y = red) \mathbb{P}(Y = red)}{\mathbb{P}(Z \in \mathcal{R})}$$
$$= k(\mathbb{P}(Y^{(1)} = blue, | Y = blue, Z \in \mathcal{R}) \mathbb{P}(Z \in \mathcal{R} | Y = blue) -$$
$$\mathbb{P}(Y^{(1)} = blue, | Y = red, Z \in \mathcal{R}) \mathbb{P}(Z \in \mathcal{R} | Y = red)),$$

where $k = \frac{1}{2\mathbb{P}(Z \in \mathcal{R})} > 0$.

Now,[5]

$$\mathbb{P}(Y^{(1)} = blue | Y = blue, Z \in \mathcal{R}) = \frac{\int_{\mathcal{R}} \mathbb{P}(Y^{(1)} = blue | Y = blue, Z = z) f_{Z|Y}(z|blue) dz}{\mathbb{P}(Z \in \mathcal{R} | Y = blue)}.$$

Thus, we have:

$$\Delta Acc_{\mathcal{R}} = k \int_{\mathcal{R}} \mathbb{P}(Y^{(1)} = blue | Y = blue, Z = z) f_{Z|Y}(z|blue) dz -$$
$$k \int_{\mathcal{R}} \mathbb{P}(Y^{(1)} = blue | Y = red, Z = z) f_{Z|Y}(z|red) dz.$$

Let us observe that, for every $z \in \mathcal{R}$, $\mathbb{P}(Y^{(1)} = blue | Y = blue, Z = z)$ can be expressed as follows:

$$\mathbb{P}(Y^{(1)} = blue | Y = blue, Z = z) = \mathbb{P}(\exists r > 0, \epsilon > 0 \text{ such that there are no training points in } [z - r, z + r],$$
$$\text{there is at least a blue point in} [z - r - \epsilon, z - r] \text{ or in } [z + r, z + r + \epsilon],$$
$$\text{there are no red points in } [z - r - \epsilon, z - r] \text{ and in } [z + r, z + r + \epsilon]).$$

_______________

[5] For the subsequent notation, refer to Bertsekas & Tsitsiklis (2008).

Similar considerations can be applied to $\mathbb{P}(Y^{(1)} = blue|Y = red, Z = z)$. Therefore, they are independent of the value of $Y$ and thus:

$$\mathbb{P}(Y^{(1)} = blue|Y = blue, Z = z) = \mathbb{P}(Y^{(1)} = blue|Y = red, Z = z).$$

Hence:

$$\Delta Acc_{\mathcal{R}} = k \int_{\mathcal{R}} \mathbb{P}(Y^{(1)} = blue|Y = blue, Z = z)(f_{Z|Y}(z|blue) - f_{Z|Y}(z|red))dz.$$

Since $f_{Z|Y}(z|blue) = 2(1 - z) > f_{Z|Y}(z|red) = 2z$ for each $z \in \mathcal{R}$, it follows that $\Delta Acc_{\mathcal{R}} \geq 0$.

### 7.3 ADDITIONAL CONSIDERATIONS

Note that the previous reasoning remains valid when considering any subset $\mathcal{B} \subseteq \mathcal{R}$. If we examine an interval of the same width on the opposite side of the neighborhood around $z^*$, it is reasonable to expect similar results, thereby suggesting that the proposed method is likely to maintain or potentially enhance the classification performance of a pre-trained neural network across an entire neighborhood of its decision boundary. This expectation arises from the fact that, on the right side of $z^*$, blue points are likely to be classified as blue by the hybrid model, given that their nearest neighbors are generally also classified as blue in this region. Consequently, our hybrid model may assist in correcting the misclassification of red points within this region. A more formal analysis of this result will be addressed in future work.

## 8 FURHTER SAMPLES FOR NNS

We show further samples, where NN impacted the prediction (see Figures 5 and 6).

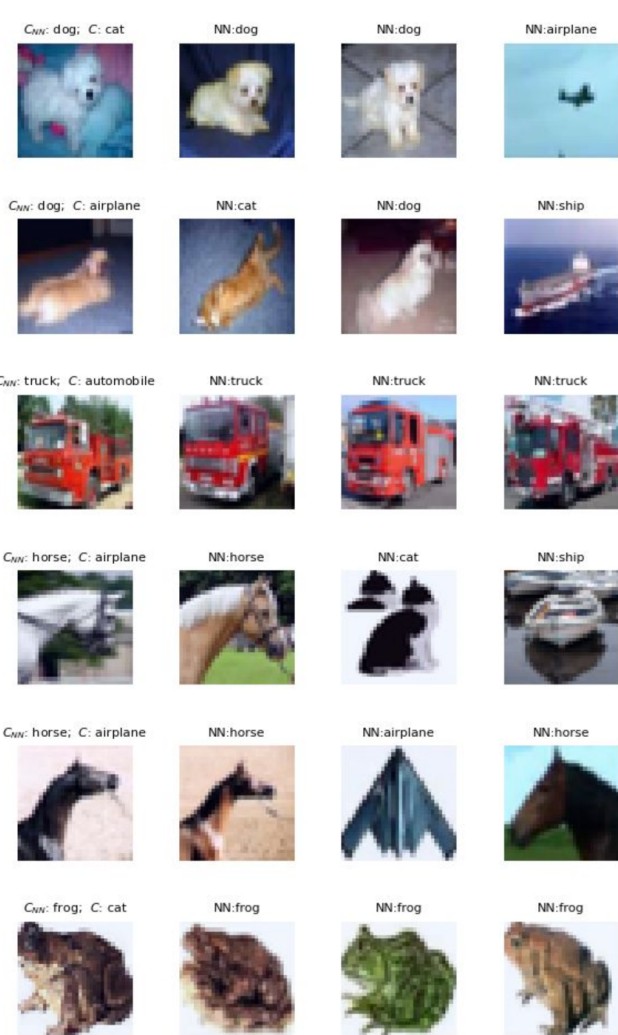

Figure 5: The first column shows samples where NNs (columns 2 to 4) yielded the correct prediction but without NNs the prediction was incorrect.

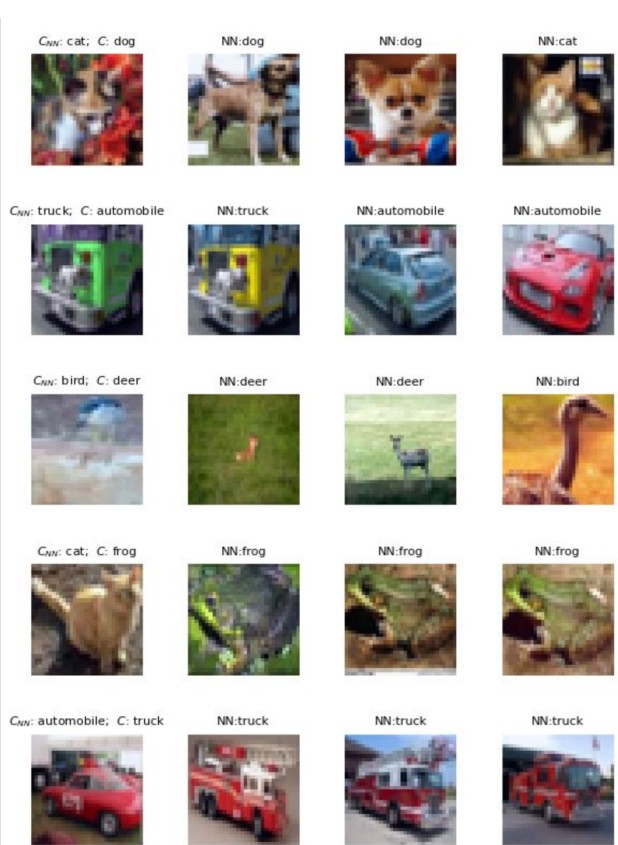

Figure 6: The first column shows samples where NNs (columns 2 to 4) yielded the incorrect prediction but without NN the prediction was correct.

