# OpenReview forum: "Improving classifier decision boundaries and interpretability using nearest neighbors"
_ICLR.cc/2025/Conference — ICLR 2025 Conference Withdrawn Submission_

### Official Review · Reviewer_Zyhd · 2024-10-27

**Soundness:** 2
**Presentation:** 3
**Contribution:** 2
**Rating:** 3
**Confidence:** 5

**Summary:**

The paper Improving Classifier Decision Boundaries and Interpretability Using Nearest Neighbors introduces LaSeNN, a method that combines neural network predictions with latent space k-nearest neighbors to smooth decision boundaries. It enhances accuracy, robustness to noise and attacks, and interpretability without modifying architectures or training procedures. Evaluated on CIFAR-10, CIFAR-100, and ImageNet, the approach offers consistent, though modest, performance gains, especially in sparse data regions.

**Strengths:**

* Enhances resistance to label noise and adversarial attacks.
* Offers insights into model decisions by leveraging nearest neighbors.
* Works with pre-trained and self-trained networks without requiring architecture changes.
* Achieves multiple objectives (accuracy, robustness, interpretability) simultaneously.
* Performs consistently across datasets (CIFAR-10/100, ImageNet) and architectures (VGG, ResNet).

**Weaknesses:**

Major Issues:
* Lack of Comparison to Baselines: The paper does not compare results with other baselines, including adversarially trained models and test-time methods, despite proposing a test-time method like [1].
* No Targeted Attack Evaluation: A targeted attack specifically designed for their method is missing. I recommend testing an attack that aims to maximize the latent space distance from the sample in the chosen latent space used for KNN computation.
* No Runtime Analysis: The paper lacks a runtime comparison between their method, the base classifier, and other baselines.

Minor Issues:
* Use of AutoAttack: AutoAttack [2] should be used instead of the attacks implemented in the paper.
* Limited Improvement: The reported performance gains are relatively small.

References:
[1] Blau, Tsachi, et al. "Classifier robustness enhancement via test-time transformation." *arXiv preprint* arXiv:2303.15409 (2023).
[2] Croce, Francesco, and Matthias Hein. "Reliable evaluation of adversarial robustness with an ensemble of diverse parameter-free attacks." *International Conference on Machine Learning*. PMLR, 2020.

**Questions:**

see weakness

---

### Official Review · Reviewer_CtCf · 2024-10-30

**Soundness:** 1
**Presentation:** 3
**Contribution:** 2
**Rating:** 3
**Confidence:** 5

**Summary:**

The paper presents LAtent-SElf-kNN (LaSeNN), a novel algorithm that improves neural network classification by combining predictions from a network with predictions from k-nearest neighbors in latent space. The authors state that neural networks learn suboptimal decision boundaries in areas of low training data density, which can lead to overfitting since these boundaries are influenced by few samples. To address this, LaSeNN
1. Computes nearest neighbors using layer activations from a pre-trained network
2. Makes predictions using a weighted average between the original network output (weight ~75%) and the predictions of nearest neighbors (weight ~25%)
3. Does not require any modifications to network architecture or training

The authors state their method shows improvements in four areas:

Classification accuracy (0.1-3% gains)
Label noise robustness (increasing gains with higher noise levels)
Adversarial attack resistance
Model interpretability through analysis of nearest neighbors

The authors validate their approach on both self-trained and pre-trained networks (e.g. VGG-13, ResNet-10) using CIFAR-10/100 and ImageNet datasets. They aim to provide theoretical analysis showing that their method maintains or improves classification performance in regions near decision boundaries.

**Strengths:**

The paper presents a novel approach to improving neural network classification by combining network predictions with k-nearest neighbors in latent space (LaSeNN). The idea of using nearest neighbors in latent space rather than input space or final outputs is innovative. The approach uniquely addresses multiple challenges (accuracy, robustness, interpretability) simultaneously without requiring architectural changes.

The evaluation is comprehensive across multiple dimensions, including testing on various architectures (VGG, ResNet, MobileNetv3, ConvNext), different datasets (CIFAR-10/100, ImageNet), and both self-trained and pre-trained networks. The authors provide a thorough analysis of parameter sensitivity, examining similarity metrics, number of neighbors, and layer selection. The work is supported by a strong theoretical foundation with mathematical proof of performance guarantees and rigorous empirical validation of assumptions about latent space distributions.

The paper presents a clear problem motivation and explains limitations in current neural networks. The methodology is well-structured with detailed algorithm descriptions, and concepts are effectively illustrated through visualizations (Figures 2-4). The experimental results are systematically organized with detailed tables and analysis, making the findings easily accessible to readers.

The work aims to demonstrate improvements in multiple critical areas: classification accuracy, robustness to label noise, adversarial attack resistance, and model interpretability. Its practical applicability to existing pre-trained models without modification would make it particularly valuable. The paper aims to contribute to the theoretical understanding of neural network decision boundaries and provide insights into the relationship between data density and classification performance.

The paper aims to provide a simple yet effective approach that addresses multiple challenges simultaneously while maintaining practical applicability and theoretical soundness.

**Weaknesses:**

The improvements in classification accuracy are quite modest, ranging from 0.1% to 3%. These incremental improvements may not justify the additional  overhead of computing nearest neighbors.

The paper acknowledges that "obtaining kNNs is also computationally expensive" but does not quantify this overhead. An analysis of computational costs and memory requirements would improve the paper, especially for large-scale deployment. The method requires storing and computing activations. The paper could benefit from discussing such practical aspects, since the work presents itself as a practical application.

The theoretical analysis only proves that the method "is expected not to diminish classification performance" in a specific region R under simplified assumptions (binary classification - blue, red). The proof does not extend to the general case, limiting its practical implications.

While the method shows improved robustness to adversarial attacks, the gains seem too low for practical use.

The optimal choice of layer i for computing nearest neighbors varies significantly between architectures and datasets (Table 4). The paper could provide guidelines for selecting this hyperparameters.

While the method provides some interpretability through nearest neighbors, the analysis in Section 3.6 is largely qualitative and anecdotal.

Generally, points raised in abstract are not addressed.

**Questions:**

To list a few of many:
1. The tables provided in Figure 2 apply to CIFAR-10 on the left (10 classes) and Imagenet on the right (1000 classes). Two classes are compared - 3 and 5 - CIFAR-10, and another two - 5 and 6 - Imagenet, respectively. How can the authors claim the figure represents a general case ? Where are the results ?
2. CIFAR-10/100 images are 32x32 originally (line 195), no scaling required ?
3. The work of Zhang et a. 2021 is not related to "erratic decision boundaries", as implied in line 494 - how was the inference made ?
4. Acronym TCAV not defined - line 368 - what does it mean ?
5. There is no mention of "Basic Iterative Attack(BIA)" in Madry et al., 2018. Where did BIA come from ?
6. .zip file gives no details of how your algorithm was implemented. Implementation details in general are poor. Are there any further details available ?
7. There is no mention of PGD in Kurakin et al., 2016 ?

---

### Official Review · Reviewer_Tm5Y · 2024-11-01

**Soundness:** 2
**Presentation:** 2
**Contribution:** 1
**Rating:** 3
**Confidence:** 4

**Summary:**

The paper proposes using nearest neighbors in combination with a neural network for image classification due to a multitude of reasons including improved robustness, explainability, and the smoothness of decision boundary. In particular, the authors proposed computing nearest neighbors using an intermediate model representation, and feeding the retrieved nearest neighbors to the classifier in addition to the original input example. The final prediction is computed as a convex combination of the prediction on the original example, and the predictions on the retrieved nearest neighbor examples. The authors also evaluated their approach under the presence of label noise and adversarial attacks.

**Strengths:**

- The proposed approach is simple and builds on a large body of work in this space
- The authors considered a range of different applications
- The paper is simple to read

**Weaknesses:**

- The formulation is almost identical to classical kNN language models (kNN-LM) -- even though they are defined for language, next-token prediction is still a classification task. The authors highlighted the differences in terms of approach in the related work section by mentioning that they don't leverage the ground-truth labels. However, I would have preferred for them to use those ground truth labels in comparison to rerunning the model on the retrieved nearest neighbors. Therefore, the reasons for this minor deviation in approach are unclear, and even if they are justified, this doesn't count as a significant novelty in terms of the method.
- While the methodology isn't particularly novel, the results are even weaker (the prediction changes perhaps for only a couple of examples in the dataset). Hence, I find it hard to justify the contribution of the paper from either the method or the results standpoint.

**Questions:**

- I would suggest the authors to use kNN instead of NN as NN also stands for `Neural Network`.
- Fig 4. (row 4): all the plotted nearest neighbors are from the `truck` class, while the suggested C_NN prediction is `automobile`. Perhaps just the kNN and CNN predictions are just flipped? Same with row 3.
- I would be happy to change my score if the authors can show a significant improvement in terms of results even if the method isn't particularly novel

---

### Official Review · Reviewer_AwSM · 2024-11-04

**Soundness:** 2
**Presentation:** 2
**Contribution:** 1
**Rating:** 3
**Confidence:** 5

**Summary:**

This paper introduces LaSeNN, a method that improves neural network robustness by mitigating issues with decision boundaries in low-density training regions. By averaging predictions with those of nearest neighbors in latent space, LaSeNN enhances label noise resistance, adversarial robustness, classification accuracy, and interpretability without altering network architecture or training methods. The approach demonstrates effectiveness across various convolutional neural networks, offering a straightforward alternative to existing, often complex solutions.

**Strengths:**

1. Focused on trustworthiness: The paper addresses an important challenge in trustworthy machine learning, a valuable try given the iusses in real-world data.
2. Dataset and models Variety in Evaluation: The authors evaluate LaSeNN on both CIFAR-10 and CIFAR-100, as well as several popular models, which showcase their adaptability across datasets of varying complexity.

**Weaknesses:**

1. Presentation Issues: Several typographical and presentation inconsistencies are present, notably the figure at the top of Figure 2(a), which appears to be a typographical error. Such issues detract from the overall clarity of the paper.

2. Questionable Novelty: The novelty of the proposed approach is unclear, as the idea shares similarities with existing work such as *Deep k-Nearest Neighbors: Towards Confident, Interpretable and Robust Deep Learning*. A more explicit differentiation is needed to clarify the unique contributions of LaSeNN relative to prior approaches.

3. Interpretation of Accuracy Values: Across all tables, the values do not appear to represent accuracy percentages, yet this is not clearly stated. Additionally, in Table 6, LaSeNN’s performance shows limited variation in accuracy as noise increases, which is unexpected. Further exploration of this trend, alongside comparisons with advanced label noise-handling methods, would strengthen the analysis.

4. Inconsistent Dataset Performance: Table 7 indicates that LaSeNN shows negligible accuracy change on CIFAR-10 even with varying k-values, while the impact is more pronounced on CIFAR-100. Similarly, in Table 8, accuracy shifts minimally for ResNet-10 on both CIFAR-10 and CIFAR-100, but significantly for VGG13 on the same datasets. These disparities necessitate a deeper discussion on why performance varies so distinctly by dataset and model.

5. Limited Comparison with Advanced Methods: The comparison baseline could be expanded to include more advanced techniques designed for managing label noise, providing a more comprehensive assessment of LaSeNN’s efficacy.

**Questions:**

1.  Can you clarify the differences in novelty between this work and the *Deep k-Nearest Neighbors* approach to better establish LaSeNN’s unique contributions?

2. In Tables 6 and 7, why does LaSeNN exhibit such minimal accuracy change with noise and varying k-values on CIFAR-10 but more noticeable changes on CIFAR-100?

3. Similarly, what accounts for the distinct accuracy changes between ResNet-10 and VGG13 on CIFAR-10 and CIFAR-100 as seen in Table 8?

4. Could you expand the comparisons in Table 6 to include more advanced methods for handling label noise?

---

### Note · Authors · 2024-11-12

**Comment:**

Thanks for the reviews. Given the stated opinions, there is no point in discussing/clarifying, judging from experience.

**Withdrawal Confirmation:**

I have read and agree with the venue's withdrawal policy on behalf of myself and my co-authors.